# INCREMENTAL FEW-SHOT LEARNING VIA VECTOR QUANTIZATION IN DEEP EMBEDDED SPACE

**Kuilin Chen**
Department of Mechanical and Industrial Engineering
University of Toronto
Toronto, Ontario, Canada
kuilin.chen@mail.utoronto.ca

**Chi-Guhn Lee**
Department of Mechanical and Industrial Engineering
University of Toronto
Toronto, Ontario, Canada
cglee@mie.utoronto.ca

## ABSTRACT

The capability of incrementally learning new tasks without forgetting old ones is a challenging problem due to catastrophic forgetting. This challenge becomes greater when novel tasks contain very few labelled training samples. Currently, most methods are dedicated to class-incremental learning and rely on sufficient training data to learn additional weights for newly added classes. Those methods cannot be easily extended to incremental regression tasks and could suffer from severe overfitting when learning few-shot novel tasks. In this study, we propose a nonparametric method in deep embedded space to tackle incremental few-shot learning problems. The knowledge about the learned tasks is compressed into a small number of quantized reference vectors. The proposed method learns new tasks sequentially by adding more reference vectors to the model using few-shot samples in each novel task. For classification problems, we employ the nearest neighbor scheme to make classification on sparsely available data and incorporate intra-class variation, less forgetting regularization and calibration of reference vectors to mitigate catastrophic forgetting. In addition, the proposed learning vector quantization (LVQ) in deep embedded space can be customized as a kernel smoother to handle incremental few-shot regression tasks. Experimental results demonstrate that the proposed method outperforms other state-of-the-art methods in incremental learning.

## 1 INTRODUCTION

Incremental learning is a learning paradigm that allows the model to continually learn new tasks on novel data, without forgetting how to perform previously learned tasks (Cauwenberghs & Poggio, 2001; Kuzborskij et al., 2013; Mensink et al., 2013). The capability of incremental learning becomes more important in real-world applications, in which the deployed models are exposed to possible out-of-sample data. Typically, hundreds of thousands of labelled samples in new tasks are required to re-train or fine-tune the model (Rebuffi et al., 2017). Unfortunately, it is impractical to gather sufficient samples of new tasks in real applications. In contrast, humans can learn new concepts from just one or a few examples, without losing old knowledge. Therefore, it is desirable to develop algorithms to support incremental learning from very few samples.

While a natural approach for incremental few-shot learning is to fine-tune part of the base model using novel training data (Donahue et al., 2014; Girshick et al., 2014), the model could suffer from severe over-fitting on new tasks due to a limited number of training samples. Moreover, simple fine-tuning also leads to significant performance drop on previously learned tasks, termed as *catastrophic forgetting* (Goodfellow et al., 2014). Recent attempts to mitigate the catastrophic forgetting are generally categorized into two streams: memory relay of old training samples (Rebuffi et al., 2017; Shin et al., 2017; Kemker & Kanan, 2018) and regularization on important model parameters (Kirkpatrick et al., 2017; Zenke et al., 2017). However, those incremental learning approaches are developed and tested on unrealistic scenarios where sufficient training samples are available in novel tasks. They may not work well when the training samples in novel tasks are few (Tao et al., 2020b).

To the best of our knowledge, the majority of incremental learning methodologies focus on classification problems and they cannot be extended to regression problems easily. In class-incremental learning, the model has to expand output dimensions to learn $N'$ novel classes while keeping the knowledge of existing $N$ classes. Parametric models estimate additional classification weights for novel classes, while nonparametric methods compute the class centroids for novel classes. In comparison, output dimensions in regression problems do not change in incremental learning as neither additional weights nor class centroids are applicable to regression problems.

Besides, we find that catastrophic forgetting in incremental few-shot classification can be attributed to three reasons. First, the model is biased towards new classes and forgets old classes because the model is fine-tuned on new data only (Hou et al., 2019; Zhao et al., 2020). Meanwhile, the prediction accuracy on novel classes is not good due to over-fitting on few-shot training samples. Second, features of novel samples could overlap with those of old classes in the feature space, leading to ambiguity among classes in the feature space. Finally, features of old classes and classification weights are no longer compatible after the model is fine-tuned with new data.

In this paper, we investigate the problem of incremental few-shot learning, where only a few training samples are available in new tasks. A unified model is learned sequentially to jointly recognize all classes or regression targets that have been encountered in previous tasks (Rebuffi et al., 2017; Wu et al., 2019). To tackle aforementioned problems, we propose a nonparametric method to handle incremental few-shot learning based on learning vector quantization (LVQ) (Sato & Yamada, 1996) in deep embedded space. As such, the adverse effects of imbalanced weights in a parametric classifier can be completely avoided (Mensink et al., 2013; Snell et al., 2017; Yu et al., 2020). Our contributions are three fold. First, a unified framework is developed, termed as incremental deep learning vector quantization (IDLVQ), to handle both incremental classification (IDLVQ-C) and regression (IDLVQ-R) problems. Second, we develop intra-class variance regularization, less forgetting constraints and calibration factors to mitigate catastrophic forgetting in class-incremental learning. Finally, the proposed methods achieve state-of-the-art performance on incremental few-shot classification and regression datasets.

## 2 RELATED WORK

**Incremental learning:** Some incremental learning approaches rely on memory replay of old exemplars to prevent forgetting previously learned knowledge. Old exemplars can be saved in memory (Rebuffi et al., 2017; Castro et al., 2018; Prabhu et al., 2020) or sampled from generative models (Shin et al., 2017; Kemker & Kanan, 2018; van de Ven et al., 2020). However, explicit storage of training samples is not scalable if the number of classes is large. Furthermore, it is difficult to train a reliable generative model for all classes from very few training samples. In parallel, regularization approaches do not require old exemplars and impose regularization on network weights or outputs to minimize the change of parameters that are important to old tasks (Kirkpatrick et al., 2017; Zenke et al., 2017). To avoid quick performance deterioration after learning a sequence of novel tasks in regularization approaches, semantic drift compensation (SDC) is developed by learning an embedding network via triplet loss (Schroff et al., 2015) and compensates the drift of class centroids using novel data only (Yu et al., 2020). In comparison, IDLVQ-C saves only one exemplar per class and uses saved exemplars to regularize the change in feature extractor and calibrate the change in reference vectors.

**Few-shot learning:** Few-shot learning attempts to obtain models for classification or regression tasks with only a few labelled samples. Few-shot models are trained on widely-varying episodes of *fake* few-shot tasks with labelled samples drawn from a large-scale meta-training dataset (Vinyals et al., 2016; Finn et al., 2017; Ravi & Larochelle, 2017; Snell et al., 2017; Sung et al., 2018). Meanwhile, recent works attempt to handle novel few-shot tasks while retraining the knowledge of the base task. These methods are referred to as dynamic few-shot learning (Gidaris & Komodakis, 2018; Ren et al., 2019a; Gidaris & Komodakis, 2019). However, dynamic few-shot learning is different from incremental few-shot learning, because they rely on the entire base training dataset and an extra meta-training dataset during meta-training. In addition, dynamic few-shot learning does not accumulate knowledge for multiple novel tasks sequentially.

**Incremental few-shot learning:** Prior works on incremental few-shot learning focus on classification problems by computing the weights for novel classes in parametric classifiers, without iterative

gradient descent. For instance, the weights of novel classes can be imprinted by normalized prototypes of novel classes, while keeping the feature extractor fixed (Qi et al., 2018). Since novel weights are computed only with the samples of novel classes, the fixed feature extractor may not be compatible with novel classification weights. More recently, neural gas network is employed to construct an undirected graph to represent knowledge of old classes (Tao et al., 2020b;a). The vertices in the graph are constructed in an unsupervised manner using competitive Hebbian learning (Fritzke, 1995), while the feature embedding is fixed. In contrast, IDLVQ learns both feature extractor and reference vectors concurrently in a supervised manner.

## 3 BACKGROUND

### 3.1 INCREMENTAL FEW-SHOT LEARNING

In this paper, incremental few-shot learning is studied for both classification and regression tasks. For classification tasks, we consider the standard class-incremental setup in literature. After the model is trained on a base task ($t = 1$) with sufficient data, the model learns novel tasks sequentially. Each novel task contains a number of novel classes with only a few training samples per class. Learning a novel task ($t > 1$) is referred to as an incremental learning session. In task $t$, we have access only to training data $\mathcal{D}^t$ in the current task and previously saved exemplars (one exemplar per class in this study). Each task has a set of classes $C^t = \{c_1^t, ..., c_{n^t}^t\}$, where $n_t$ is the number of classes in task $t$. In addition, it is assumed that there is no overlap between classes in different tasks $C^t \bigcap C^s = \varnothing$ for $t \neq s$. After an incremental learning session, the performance of the model is evaluated on a test set that contains all previously seen classes $C = \bigcap_i C^i$. Note that our focus is *not* multi-task scenario, where a task ID is exposed to the model during test phase and the model is only required to perform a given task one time (van de Ven & Tolias, 2019). Our model is evaluated in a task-agnostic setting, where task ID is not exposed to the model at test time.

For regression tasks, we follow a similar setting with a notable difference that the target is real-valued $y \in \mathbb{R}$. In addition, the target values in different tasks do not have to be mutually exclusive, unlike the class-incremental setup.

### 3.2 LEARNING VECTOR QUANTIZATION

Traditional nonparametric methods, such as nearest neighbors, represent knowledge and make predictions by storing the entire training set. Despite the simplicity and effectiveness, they are not scalable to a large-scale base dataset. Typically, incremental learning methods are only allowed to store a small number of exemplars to preserve the knowledge of previously learned tasks. However, randomly selected exemplars may not well present the knowledge in old tasks. LVQ is a classical data compression method that represents the knowledge through a few learned reference vectors (Sato & Yamada, 1996; Seo & Obermayer, 2003; Biehl et al., 2007). A new sample is classified to the same label as the nearest reference vector in the input space. LVQ has been combined with deep feature extractors as an alternative to standard neural networks for better interpretability (De Vries et al., 2016; Villmann et al., 2017; Saralajew et al., 2018). The combinations of LVQ and deep feature extractors have been applied to natural language processing (NLP), facial recognition and biometrics (Variani et al., 2015; Wang et al., 2016; Ren et al., 2019b; Leng et al., 2015). We notice that LVQ is a nonparametric method which is well suited for incremental few-shot learning because the model capacity grows by incorporating more reference vectors to learn new knowledge. For example, incremental learning vector quantization (ILVQ) has been developed to learn classification models adaptively from raw features (Xu et al., 2012). In this study, we present the knowledge by learning reference vectors in the feature space through LVQ and adapt them in incremental few-shot learning. Compared with ILVQ by Xu et al. (2012), our method does not rely on predefined rules to update reference vectors and can be learned along with deep neural networks in an end-to-end fashion. Besides, our method uses a single reference vector for each class, while ILVQ automatically assigns different numbers of prototypes for different classes.

## 4 METHODOLOGY

### 4.1 INCREMENTAL DEEP LEARNING VECTOR QUANTIZATION

The general framework of IDLVQ for both classification and regression can be derived from a Gaussian mixture perspective (Ghahramani & Jordan, 1994), with a simplified covariance structure and supervised deep representation learning. In the base dataset ($t = 1$), a raw input $\mathbf{x}$ is projected into a feature space $\mathcal{F}^1$ by a deep neural network $f_{\theta^1}$, where $\theta^1$ denotes the parameters in neural networks. In addition, $N^1$ reference vectors $\mathbf{M}^1 = \{\mathbf{m}_1^1, ..., \mathbf{m}_{N^1}^1\}$ are placed in the feature space $\mathcal{F}^1$, which can be learned to capture the representation of the base dataset. More reference vectors will be added incrementally while learning novel tasks. The marginal distribution $p(f_{\theta^1}(\mathbf{x}))$ of feature vector can be described by a Gaussian mixture model $p(f_{\theta^1}(\mathbf{x})) = \sum_{i=1}^{N^1} p(i)p(f_{\theta^1}(\mathbf{x})|i)$ of $N^1$ components, where the prior $p(i) = 1/N^1$ and the component distribution $p(f_{\theta^1}(\mathbf{x})|i)$ is Gaussian. By assuming that each component distribution $p(f_{\theta^1}(\mathbf{x})|i)$ is isotropic Gaussian centered at $\mathbf{m}_i^1$ with the same covariance, the posterior distribution of a component given the input is

$$p^1(i|\mathbf{x}) = \frac{\kappa(f_{\theta^1}(\mathbf{x}), \mathbf{m}_i^1)}{\sum_{j=1}^{N^1} \kappa(f_{\theta^1}(\mathbf{x}), \mathbf{m}_j^1)}, \tag{1}$$

where $\kappa(f_{\theta^1}(\mathbf{x}), \mathbf{m}_i^1) = \exp(-\|f_{\theta^1}(\mathbf{x}) - \mathbf{m}_i^1\|^2/\gamma)$ is a Gaussian kernel and $\gamma$ is a scale factor. The conditional expectation of the output from a Gaussian mixture is $\hat{y} = \sum_{i=1}^{N^1} p^1(i|\mathbf{x})q_i^1$, where $q_i^1$ is the reference target associated with reference vector $\mathbf{m}_i^1$. In classification problems, $q_i^1$ is either 0 or 1 indicating whether $\mathbf{m}_i^1$ and $\mathbf{x}$ have the same label. Since each reference vector is assigned to a class at initialization, $q_i^1$ is fixed and does not require learning. Meanwhile, $q_i^1$ in regression problems is real-valued and has to be learned. The weights in neural networks $\theta^1$, reference vectors $\mathbf{M}^1$, reference targets $q_i^1$ (in regression problems only) and the scale factor $\gamma$ are learned concurrently by minimizing a loss function between the true label $y$ and the predicted label $\hat{y}$.

The proposed IDLVQ is a nonparametric method as it makes prediction based on similarity to reference vectors, instead of using any regression or classification weights. The capacity of the model grows naturally by adding more reference vectors to learn novel tasks, while the old knowledge is preserved in existing reference vectors.

### 4.2 INCREMENTAL DEEP LEARNING VECTOR QUANTIZATION FOR CLASSIFICATION

For classification problems, one reference vector is assigned to each class in our study. Thus, $\hat{y}$ represents the predicted probability that an input belongs to a class. The model can be trained to classify data correctly by minimizing the cross-entropy loss $\mathcal{L}_{CE}$ between the predicted probability $\hat{y}$ and the true label $y$. Although the cross-entropy loss encourages separability of features in base classes, it does not guarantee compact intra-class variation in the feature space. Specifically, in an incremental learning session, features of novel classes could overlap with those of previously learned classes. As a result, the overall classification accuracy could deteriorate after incremental learning sessions. A desirable feature embedding leaves large margin between classes to mitigate overlap in features across old and new classes. Inspired by center loss (Wen et al., 2016) to enhance discriminative capability in facial recognition, a regularization term on intra-class distance to reference vectors is added to get compact intra-class variation.

$$\mathcal{L}_{intra} = \sum_{\forall(\mathbf{x},y), y=i} \left\| f_{\theta^1}(\mathbf{x}) - \mathbf{m}_i^1 \right\|^2 \tag{2}$$

As such, $f_{\theta^1}(\mathbf{x})$ is forced to stay close to the reference vector with the same label and naturally moves away from other reference vectors. Consequently, features of new classes are more likely to lie in the margin between old classes to mitigate ambiguity in features across different classes. The total loss in training the base task is given by $\mathcal{L} = \mathcal{L}_{CE} + \lambda_{intra}\mathcal{L}_{intra}$, where $\lambda_{intra}$ is a hyper-parameter to control the weight for intra-class variation loss. The total loss is differentiable w.r.t. neural network parameters $\theta^1$, reference vectors $\mathbf{M}^1 = \{\mathbf{m}_1^1, ..., \mathbf{m}_{n^1}^1\}$ and scaling factor $\gamma$. All parameters in the model can be trained jointly in an end-to-end fashion.

In an incremental session ($t > 1$), a novel dataset $\mathcal{D}^t$ contains $n^t$ classes and $K^t$ samples per class ($n^t$-way $K^t$-shot). $n^t$ new reference vectors are added and each reference vector is initialized

as the centroid of features in a class $\mathbf{m}_i^t = \frac{1}{K^t} \sum_{k=1}^{K^t} f_{\theta^t}(\mathbf{x}_k)$. The new reference vectors along with the neural network parameters are fine-tuned on $\mathcal{D}^t$ to learn new knowledge in task $t$. To preserve the knowledge from the old tasks during incremental learning, the model should be updated only when necessary. Therefore, cross-entropy loss is not used in incremental learning sessions because it always updates model parameters even if the sample is correctly classified. Let $\mathbf{m}_+^t$ be the reference vector with the correct label and $\mathbf{m}_-^t$ be the nearest reference vector with a wrong label. For a training sample $(\mathbf{x}, y)$ in $\mathcal{D}^t$, the sample is classified correctly if $\left\| f_{\theta^t}(\mathbf{x}) - \mathbf{m}_+^t \right\|^2 < \left\| f_{\theta^t}(\mathbf{x}) - \mathbf{m}_-^t \right\|^2$. In this case, the loss should be 0. When $\left\| f_{\theta^t}(\mathbf{x}) - \mathbf{m}_+^t \right\|^2 > \left\| f_{\theta^t}(\mathbf{x}) - \mathbf{m}_-^t \right\|^2$, the sample is misclassified. We adapt the margin based loss function $\mathcal{L}_M$ from De Vries et al. (2016) with a minor modification

$$\mathcal{L}_M = \text{ReLU} \left( \frac{\left\| f_{\theta^t}(\mathbf{x}) - \mathbf{m}_+^t \right\|^2 - \left\| f_{\theta^t}(\mathbf{x}) - \mathbf{m}_-^t \right\|^2}{\left\| f_{\theta^t}(\mathbf{x}) - \mathbf{m}_+^t \right\|^2 + \left\| f_{\theta^t}(\mathbf{x}) - \mathbf{m}_-^t \right\|^2} \right), \quad (3)$$

where $\text{ReLU}(\cdot)$ stands for the rectified linear unit function. The margin based loss leads to slow training convergence because it only updates two reference vectors one time. However, the adapted margin based loss is well suited in learning from few-shot samples while avoids unnecessary parameter updates.

Features for an old class could deviate away from the corresponding reference vector due to changes in $\theta^t$ during incremental learning, leading to catastrophic forgetting. A forgetting loss $\mathcal{L}_F$ is developed to regularize the drift in the feature space

$$\mathcal{L}_F = \sum_{i=1}^{N^{t-1}} \| f_{\theta^t}(\mathbf{x}_i^{'}) - f_{\theta^{t-1}}(\mathbf{x}_i^{'}) \|^2, \quad (4)$$

where $\mathbf{x}_i^{'}$ is the selected exemplar for class $i$ and $N^{t-1}$ denotes the total number of classes in the base task and all previous novel tasks. Note that the exemplar $\mathbf{x}_i^{'}$ for class $i \in [N^{t-1}, N^t]$ is picked from $\mathcal{D}^t$ whose feature is nearest to $\mathbf{m}_i^t$ at the end of each learning session. The total loss in the incremental learning session $t$ is $\mathcal{L} = \mathcal{L}_M + \lambda_F \mathcal{L}_F + \lambda_{intra} \mathcal{L}_{intra}$, where $\lambda_F$ and $\lambda_{intra}$ are weights for forgetting loss and intra-class variation loss, respectively. The total loss is optimized w.r.t. neural network parameters $\theta^t$ and new reference vectors $\{\mathbf{m}_{N^{t-1}+1}^t, ..., \mathbf{m}_{N^t}^t\}$.

The reference vectors for previously learned tasks are not updated by novel data to prevent catastrophic forgetting. However, they may not be well suited to represent knowledge and make classification in the new feature space $\mathcal{F}^t$ as feature embedding is changed with updated $\theta^t$. Although the true optimal location of those reference vectors are difficult to estimate without using the entire data from all tasks, they can be calculated approximately using the shift in features of exemplars. Considering that features of an exemplar $\mathbf{x}_i^{'}$ are close to $\mathbf{m}_i$ in the feature space, the shift of a reference vector $\delta_i^t$ in the new feature space can be approximated by the shift of the exemplar's features $\delta_i^t = f_{\theta^t}(\mathbf{x}_i^{'}) - f_{\theta^{t-1}}(\mathbf{x}_i^{'})$. Therefore, the reference vectors for previously learned tasks are calibrated $\mathbf{m}_i^t = \mathbf{m}_i^{t-1} + \delta_i^t$, where $\mathbf{m}_i^{t-1}$ is the uncalibrated reference vector for class $i \in [1, N^{t-1}]$. A test sample, which could be from any seen classes, is classified according to the distance to reference vectors $\{\mathbf{m}_1^t, ..., \mathbf{m}_{N^t}^t\}$. The pseudo code for IDLVQ-C is presented in the appendix.

## 4.3 INCREMENTAL DEEP LEARNING VECTOR QUANTIZATION FOR REGRESSION

For regression problems, the model is trained to recognize regression targets by the minimizing mean squared error (MSE) loss $\mathcal{L}_{MSE} = (y - \hat{y})^2$, where $y$ is the real-valued target in training dataset. The MSE loss function is differentiable w.r.t. neural network weights, reference vectors and targets, and scale factor. Therefore, all parameters can be trained jointly in an end-to-end manner. The proposed IDLVQ-R can also be interpreted as a kernel smoother in deep embedded space. Compared with traditional kernel smoother, such as Nadaraya-Watson estimator (Nadaraya, 1964), IDLVQ-R is sparse and hence more scalable as it only relies on a few reference vectors and targets.

In an incremental learning session ($t > 1$), we have access to data $\mathcal{D}^t$ that contains $K^t$ pairs of training samples $(\mathbf{x}_i^t, y_i^t)$. $n^t$ new reference vectors ($n^t \leq K^t$) along with corresponding targets are added to the model to learn new knowledge in the novel task $t$. We randomly select $n^t$ samples from

$\mathcal{D}^t$ to initialize reference vectors and targets as follows

$$\mathbf{m}_{i+N^{t-1}} = f_\theta(\mathbf{x}_i^t), \tag{5}$$

$$q_{i+N^{t-1}} = y_i^t, \tag{6}$$

where $N^{t-1}$ is the total number of reference vectors in all previous tasks. The new reference vectors and targets are fine-tuned by minimizing MSE on $\mathcal{D}^t$ while keeping other parameters frozen. After new reference vectors and targets are fine-tuned with novel data $\mathcal{D}^t$, the model makes prediction by smoothing targets of all reference vectors

$$\hat{y} = \frac{\sum_{i=1}^{N^t} \kappa(f_\theta(\mathbf{x}), \mathbf{m}_i) q_i}{\sum_{i=1}^{N^t} \kappa(f_\theta(\mathbf{x}), \mathbf{m}_i)}, \tag{7}$$

where $N^t$ is the current total number of reference vectors.

## 5 EXPERIMENTS

We first describe the overall protocols, then we present the results on incremental few-shot classification and regression problems.

### 5.1 INCREMENTAL FEW-SHOT CLASSIFICATION

We empirically evaluate the performance of IDLVQ-C on incremental few-shot classification on CUB200-2011 (Welinder et al., 2010) and *mini*ImageNet datasets (Vinyals et al., 2016). The dataset is split into base classes and multiple groups of novel classes. We apply standard data augmentation, including random crop, horizontal flip and color jitter, on all training images. After each training session, the model performance is evaluated on a test set, which contains all classes that the model has been trained on.

**CUB** dataset is composed of 200 fine-grained bird species with 11,788 images. We split the dataset into 5894 training images, 2947 validation images and 2947 test images. All images are resized to $224 \times 224$. In addition, the first 100 classes are chosen as base classes, where all training samples in base classes are used to train the base model. The remaining 100 classes are treated as novel categories and split into 10 incremental learning sessions. Each incremental learning session contains 10 novel classes and 5 randomly selected training samples per class (10-way 5-shot).

***mini*ImageNet** dataset is a 100-class subset of the original ImageNet dataset (Deng et al., 2009). Each class contains 500 training images, 50 validation images, and 50 test images. The images are in RGB format of the size $84 \times 84$. We choose 60 and 40 classes for base and novel classes, respectfully. The 40 novel classes are divided into 8 sessions and each session contains 5 novel classes with 5 randomly selected training samples per class (5-way 5-shot).

ResNet18 (He et al., 2016) is used as the feature extractor for incremental classification problems. The learning process for each dataset is repeated 10 times and the average test accuracy is reported. The proposed method is compared with six methods for few-shot class-incremental learning: fine-turning using $\mathcal{D}^t$, joint training using the entire training set from all encountered classes, iCaRL (Rebuffi et al., 2017), Rebalancing (Hou et al., 2019), ProtoNet (Snell et al., 2017), incremental learning vector quantization (ILVQ) (Xu et al., 2012), SDC (Yu et al., 2020), and Imprint (Qi et al., 2018). Note that ILVQ is applied to the features extracted by neural networks in our experiment.

The incremental few-shot learning results on CUB and miniImageNet are shown in Table 1 and 2, respectively. Our method outperforms fine-tuning, iCaRL (Rebuffi et al., 2017), and ProtoNet (Snell et al., 2017) by a large margin. Simply fine-tuning the weights in classifier with few-shot training samples for novel classes significantly deteriorates the prediction accuracy. Although iCaRL alleviates catastrophic forgetting by tuning the model with a mix of old exemplars and novel few-shot data, the prediction accuracy still drops quickly because iCaRL requires sufficient samples per class to achieve satisfactory performance. The ProtoNet relies on distance to prototypes (the mean of features within a class) to make classification but the fixed feature extractor may not be able to well separate novel classes. ILVQ is slightly better than ProtoNet because prototypes can be learned adaptively when more classes are available in incremental learning sessions. Some prototypes in

ILVQ are close to the border of a class, which are more effective than class centroids in ProtoNet. However, ILVQ does not achieve the best performance because the feature extractor is fixed and cannot be learned along with the prototypes. IDLVQ-C has a small gain in the first couple of incremental few-shot learning sessions compared with SDC (Yu et al., 2020) and Imprint (Qi et al., 2018). Similar to ProtoNet, SDC also relies on prototypes to make classification. The performance of SDC is better than that of ProtoNet because SDC fine-tunes the feature extractor with novel dataset and compensates the drift in prototypes. However, the compensation for the drift of old-class prototypes can be less accurate in SDC because it is approximated by samples in novel classes. In parallel, the imprint method directly computes the normalized classification weights from the average of normalized features within a novel class. The imprint method avoids imbalanced classification weights and circumvents the overfitting in few-shot class-incremental learning through weight normalization. Nevertheless, the fixed feature extractor in the imprint method may not be well suited for novel classes. In contrast, IDLVQ-C updates the feature extractor only when necessary and compensates the shift of old reference vectors more accurately using exemplars from old classes. That is why the gain of IDLVQ-C increases with more incremental few-shot learning sessions. The performance of SDC, Imprint and IDLVQ-C is better than offline joint training in early sessions of incremental few-shot learning. Offline joint training may not result in oracle performance due to extremely imbalanced samples between base classes and novel classes.

Table 1: Prediction accuracy on CUB all classes using the 10-way 5-shot incremental setting.

| Method | sessions | | | | | | | | | | |
|---|---|---|---|---|---|---|---|---|---|---|---|
| | 1 | 2 | 3 | 4 | 5 | 6 | 7 | 8 | 9 | 10 | 11 |
| Fine-tune | 77.30 | 46.23 | 34.71 | 25.35 | 23.16 | 20.65 | 16.21 | 13.32 | 11.98 | 11.17 | 10.76 |
| Joint train | 77.30 | 73.28 | 68.80 | 65.34 | 63.75 | 62.00 | 60.81 | 59.71 | **59.06** | **58.69** | **58.23** |
| iCaRL (Rebuffi et al., 2017) | 77.30 | 57.18 | 54.67 | 48.11 | 40.76 | 36.85 | 33.12 | 30.42 | 28.22 | 26.84 | 25.23 |
| Rebalancing (Hou et al., 2019) | 77.30 | 64.53 | 56.14 | 47.29 | 38.92 | 34.39 | 31.04 | 27.93 | 27.12 | 24.46 | 23.61 |
| ProtoNet (Snell et al., 2017) | 77.30 | 69.76 | 66.01 | 62.29 | 59.58 | 57.10 | 55.13 | 54.09 | 52.40 | 51.65 | 50.36 |
| ILVQ (Xu et al., 2012) | 77.30 | 71.50 | 66.79 | 62.71 | 60.20 | 57.84 | 55.27 | 55.06 | 52.42 | 51.72 | 50.47 |
| SDC (Yu et al., 2020) | 77.34 | 74.45 | 69.45 | 65.27 | 61.81 | 58.26 | 56.14 | 55.71 | 53.31 | 52.79 | 51.52 |
| Imprint (Qi et al., 2018) | 77.02 | 73.39 | 69.50 | 65.61 | 62.81 | 60.74 | 59.39 | 58.61 | 56.85 | 55.93 | 54.82 |
| **IDLVQ-C** | **77.37** | **74.72** | **70.28** | **67.13** | **65.34** | **63.52** | **62.10** | **61.54** | **59.04** | **58.68** | **57.81** |

Table 2: Prediction accuracy on *mini*ImageNet all classes using the 5-way 5-shot incremental setting.

| Method | sessions | | | | | | | | |
|---|---|---|---|---|---|---|---|---|---|
| | 1 | 2 | 3 | 4 | 5 | 6 | 7 | 8 | 9 |
| Fine-tune | 64.25 | 30.11 | 18.53 | 6.31 | 2.86 | 2.68 | 1.87 | 1.56 | 1.42 |
| Joint train | 64.25 | 58.80 | 55.26 | 52.38 | 49.71 | **48.37** | **45.91** | **44.68** | **43.38** |
| iCaRL (Rebuffi et al., 2017) | 64.25 | 48.04 | 43.13 | 38.28 | 30.01 | 24.46 | 21.85 | 19.84 | 17.76 |
| Rebalancing (Hou et al., 2019) | 64.25 | 49.21 | 44.17 | 37.71 | 30.11 | 22.92 | 19.99 | 17.96 | 16.25 |
| ProtoNet (Snell et al., 2017) | 64.25 | 55.12 | 51.67 | 48.91 | 46.52 | 44.25 | 41.91 | 40.07 | 38.42 |
| ILVQ (Xu et al., 2012) | 64.25 | 56.01 | 52.43 | 49.31 | 46.98 | 44.37 | 42.06 | 40.11 | 38.43 |
| SDC (Yu et al., 2020) | 64.62 | 59.63 | 55.39 | 50.92 | 48.30 | 45.28 | 42.97 | 42.51 | 41.24 |
| Imprint (Qi et al., 2018) | 64.71 | 59.85 | 55.71 | 52.47 | **49.90** | 47.31 | 44.57 | 42.57 | 41.26 |
| **IDLVQ-C** | **64.77** | **59.87** | **55.93** | **52.62** | 49.88 | **47.55** | **44.83** | **43.14** | **41.84** |

**Ablation studies** are conducted to analyze how individual components affect the performance of incremental few-shot learning. We study five variants of our methods: (a) new reference vectors are initialized as class centroids and no tuning is done for feature extractor or old reference vectors; (b) $\mathcal{L}_{intra}$ is not used in incremental learning sessions; (c) $\mathcal{L}_F$ is not used in the incremental learning sessions; (d) shift in old reference vectors are not compensated; (e) replace the margin based loss $\mathcal{L}_M$ with the cross entropy loss $\mathcal{L}_{CE}$. Table 3 shows the results of our ablation studies on CUB dataset. Without any fine-tuning, the initial reference vectors for novel classes lead to descent accuracy in incremental few-shot classification. It demonstrates the robustness of nonparametric classifier. $\mathcal{L}_{intra}$ leads to $0.57\%$ gain due to tight intra-class variation. The less forgetting regularization $\mathcal{L}_F$ is proved to prevent forgetting old classes and achieves a performance boost by $2.35\%$. The shift compensation for reference vectors effectively adapts old reference vector in new embedding spaces with a gain round of $1\%$. The margin based loss is more effective in preventing forgetting in early sessions of incremental learning.

**The effect of the number of training samples per class.** The proposed method is evaluated under different few-shot settings on CUB dataset to investigate the effect of different numbers of training

Table 3: Ablation study on CUB using the 10-way 5-shot incremental setting.

| Method | sessions | | | | | | | | | |
|---|---|---|---|---|---|---|---|---|---|---|
| | 2 | 3 | 4 | 5 | 6 | 7 | 8 | 9 | 10 | 11 |
| No tuning | 71.93 | 67.14 | 64.21 | 62.61 | 60.13 | 59.04 | 58.47 | 55.64 | 54.25 | 53.66 |
| w.o. $\mathcal{L}_{intra}$ | 74.75 | 70.26 | 66.89 | 65.05 | 63.18 | 61.84 | 61.36 | 58.61 | 58.14 | 57.24 |
| w.o. $\mathcal{L}_F$ | 73.85 | 69.54 | 66.21 | 64.02 | 62.74 | 60.28 | 59.49 | 56.97 | 56.38 | 55.46 |
| w.o. $\delta_i$ | 74.67 | 70.01 | 66.74 | 64.81 | 63.90 | 61.42 | 60.73 | 58.16 | 57.62 | 56.79 |
| $\mathcal{L}_M \rightarrow \mathcal{L}_{CE}$ | 73.22 | 69.41 | 66.03 | 63.93 | 63.07 | 61.14 | 60.98 | 58.67 | 58.11 | 57.32 |
| IDLVQ-C | 74.72 | 70.28 | 67.13 | 65.34 | 63.52 | 62.10 | 61.54 | 59.04 | 58.68 | 57.81 |

samples in novel classes, including 5-shot, 10-shot and 20-shot settings. One reference vector is assigned to each class in all few-shot settings. As shown in Fig. 2 in the appendix, the performance of incremental learning improves as the number of samples per class increases. When the training samples are scarce, the training samples may not well present the generative distribution of training data. Therefore, the learned reference vectors could be biased and classification accuracy is low. With more training samples, the learned reference vectors could well present the center of the distribution and classification accuracy is improved. The gap in performance becomes more obvious as the number of incremental learning sessions grows. The detailed results are reported in Table 7 and 8.

## 5.2 INCREMENTAL FEW-SHOT REGRESSION

IDLVQ-R is tested on two regression datasets: sinusoidal wave and 3D spatial data. Considering that there is no state-of-the-art method for incremental few-shot regression, we compare IDLVQ-R against three alternative methods: fine-tuning using novel task data only, fine-tuning using novel task data along with exemplars and offline training using the entire training dataset from all tasks.

**Sinusoidal wave** is defined by a function $y = \sin(3\pi x) + 0.3\cos(9\pi x) + 0.5\sin(7\pi x) + \epsilon$, where $\epsilon$ is white noise with a standard deviation of 0.1. 1000 training samples in the first task (bas task) are generated by sampling $x \in [-1.0, 1.0]$ uniformly. 5-shot training samples in two novel tasks are generated by sampling $x \in [1.0, 1.5]$ and $x \in [1.5, 2.0]$, respectively.

As shown in Fig. 1(a), IDLVQ-R achieves comparable performance to offline neural networks in Fig. 1(d) which are trained using the entire training set from all tasks. In comparison, neural networks trained sequentially with few-shot training samples show catastrophic forgetting on old tasks in Fig. 1(b). With the addition of exemplars during training, the networks perform better but still suffer from catastrophic forgetting on the base task in Fig. 1(c). In conclusion, IDLVQ-R preserves old knowledge and adapts to new knowledge quickly using a few reference vectors and achieves satisfactory performance on incremental few-shot regression tasks. The experiment details can be found in the appendix.

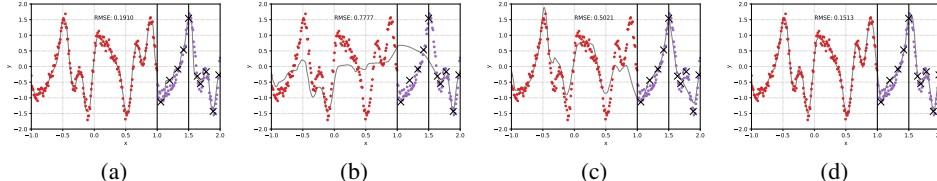

(a)  (b)  (c)  (d)

Figure 1: Comparison of performance for incremental few-shot regression. Red dots denote test samples for base task, purple dots denotes test sample for novel tasks, grey lines denote model predictions, and black crosses denotes few-shot training samples in novel tasks. (a) IDLVQ-R; (b) neural networks incrementally fine-tuned with novel data only in each session; (c) neural networks incrementally fine-tuned with exemplars and novel training samples; (d) offline neural networks trained with training samples from all tasks.

**3D spatial data**[1] is collected in North Jutland, Denmark. The inputs are longitude $x_1$ and latitude $x_2$, and the output is altitude $y$. 2482 training samples in the 1st task are collected in the area where

---

[1] https://archive.ics.uci.edu/ml/machine-learning-databases/00246/

$x_1 \in [9.98, 9.995]$ and $x_2 \in [57.0, 57.05]$. 20 training samples in the 2nd task are collected in the area where $x_1 \in [9.995, 10.0]$ and $x_2 \in [57.02, 57.03]$. 20 training samples in the 3rd task are collected in the area where $x_1 \in [9.995, 10.0]$ and $x_2 \in [57.03, 57.04]$. Approximately 300 test samples are collected for each task.

Table 4: Normalized RMSE of incremental few-shot regression on 3D spatial data

| Method | sessions | | |
|---|---|---|---|
| | 1 | 2 | 3 |
| Joint train offline | 0.02174(2e-4) | 0.02232(2e-4) | 0.02296(2e-4) |
| Fine-tune w. novel data | 0.02174(2e-4) | 0.08462(4e-4) | 0.11870(6e-4) |
| Fine-tune w. exemplars | 0.02174(2e-4) | 0.02988(2e-4) | 0.03128(2e-4) |
| IDLVQ-R | 0.02181(2e-4) | 0.02641(2e-4) | 0.02817(2e-4) |

The normalized root mean squared errors (RMSE) between actual and predicted altitude in the test set are listed in Table 4. The prediction accuracy drops significantly when the model is fine-tuned with novel data only. Catastrophic forgetting can be alleviated using exemplars from previous tasks. IDLVQ-R achieves better results than fine tuning with exemplars. The good performance of IDLVQ-R can be attributed to two reasons. First, IDLVQ-R learns a number of reference vectors and targets to preserve the knowledge in encountered tasks. Compared with a linear layer on top of neural networks, a number of reference vectors represent richer information about the training data. Second, IDLVQ-R is nonparametric and can represent local and nonlinear relationship without learning any regression coefficient from few-shot data.

## 6 CONCLUSIONS

A new incremental few-shot learning approach is developed to harmonize old knowledge preserving and new knowledge adaptation through quantized vector in deep embedded space. Prediction is made in a nonparametric way using similarity to learned reference vectors, which circumvents biased weights in a parametric classification layer during incremental few-shot learning. For classification problems, additional mechanisms are developed to mitigate the forgetting in old classes and improve representation learning for few-shot novel classes. For regression problems, the proposed approach has been reinterpreted as a kernel smoother to predict real-valued target over novel domain.

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

## A  APPENDIX

### A.1  PSEUDO CODE FOR IDLVQ-C

---
**Algorithm 1** IDLVQ-C

---
In the base task ($t = 1$)
    Initialize $\theta^1$, $\{\mathbf{m}_1^1, ..., \mathbf{m}_{N^1}^1\}$ and $\gamma$
    Minimize $\mathcal{L} = \mathcal{L}_{CE} + \lambda_{intra}\mathcal{L}_{intra}$ w.r.t. $\theta^1$, $\{\mathbf{m}_1^1, ..., \mathbf{m}_{N^1}^1\}$ and $\gamma$
    Pick exemplars from $\mathcal{D}^1$ for classes in the base task: $\mathbf{x}_i^{'} = \arg\min_{\mathbf{x}\in\mathcal{D}^1} \left\| f_{\theta^{t-1}}(\mathbf{x}) - \mathbf{m}_i^1 \right\|^2$
**for** novel task $t = 2, 3, ...$ **do**
    Initialize $\{\mathbf{m}_{N^{t-1}+1}^t, ..., \mathbf{m}_{N^t}^t\}$
    Minimize $\mathcal{L} = \mathcal{L}_M + \lambda_F\mathcal{L}_F + \lambda_{intra}\mathcal{L}_{intra}$ w.r.t. $\theta^t$ and $\{\mathbf{m}_{N^{t-1}+1}^t, ..., \mathbf{m}_{N^t}^t\}$
    Calibrate old reference vector $\{\mathbf{m}_1^{t-1}, ..., \mathbf{m}_{N^{t-1}}^{t-1}\}$ using $\mathbf{m}_i^t = \mathbf{m}_i^{t-1} + \delta_i^t$
    Pick exemplars from $\mathcal{D}^t$ for classes in the novel task $t$: $\mathbf{x}_i^{'} = \arg\min_{\mathbf{x}\in\mathcal{D}^t} \left\| f_{\theta^{t-1}}(\mathbf{x}) - \mathbf{m}_i^t \right\|^2$
**end for**

---

### A.2  PSEUDO CODE FOR IDLVQ-R

---
**Algorithm 2** IDLVQ-R

---
In the base task ($t = 1$)
    Initialize $\theta$, $\{\mathbf{m}_1^1, ..., \mathbf{m}_{N^1}^1\}$, $\{q_1^1, ..., q_{N^1}^1\}$ and $\gamma$
    Minimize $\mathcal{L}_{MSE}$ w.r.t. $\theta$, $\{\mathbf{m}_1^1, ..., \mathbf{m}_{N^1}^1\}$, $\{q_1^1, ..., q_{N^1}^1\}$ and $\gamma$
**for** novel task $t = 2, 3, ...$ **do**
    Initialize $\{\mathbf{m}_{N^{t-1}+1}^t, ..., \mathbf{m}_{N^t}^t\}$ and $\{q_{N^{t-1}+1}^t, ..., q_{N^t}^t\}$
    Minimize $\mathcal{L}_{MSE}$ w.r.t. $\{\mathbf{m}_{N^{t-1}+1}^t, ..., \mathbf{m}_{N^t}^t\}$ and $\{q_{N^{t-1}+1}^t, ..., q_{N^t}^t\}$
**end for**

---

## A.3 Experiment details for incremental few-shot classification

The base model is trained by the SGD optimizer (momentum of 0.9 and weight decay of 1e-4) with a mini-batch size of 64. For CUB dataset, the initial learning rate is 0.01 and is decayed by 0.1 after 60 and 120 epochs (200 epochs in total). For miniImageNet, the learning rate also starts from 0.01 and is decayed by 0.1 every 200 epochs (600 epochs in total). In an incremental learning session $(t > 1)$, the model is fine-tuned with with $\mathcal{D}^t$ with a learning rate of 0.01 for 100 epochs. Since novel data $\mathcal{D}^t$ $(t > 1)$ contains very few training samples, all training samples in $\mathcal{D}^t$ are included in one mini-batch. In addition, we use $\lambda_{intra} = 1.0$ and $\lambda_F = 0.5$ for both datasets. Empirically, larger $\lambda_{intra}$ leads to more compact intra-class variation. However, convergence could be slow if $\lambda_{intra}$ is too large. In addition, larger $\lambda_F$ results in less forgetting in old classes but makes learning novel classes more difficult.

## A.4 Additional results for incremental few-shot classification

The accuracies for base and novel classes are reported separately in Table 5 and 6 for CUB amd miniImageNet, respectively. The prediction accuracy of novel classes is calculated upon all novel classes the model has been trained on. Note that the accuracy in Table 1 and 2 is calculated upon all classes (including base and novel classes) that the model has been trained on. The proposed IDLVQ-C demonstrates strong capability of preserving old knowledge by achieving the best performance on old classes across all learning sessions. In parallel, Imprint method performs slightly better on novel classes than IDLVQ-C in early incremental learning sessions, while IDLVQ-C outperforms Imprint method in longer incremental learning sessions. The advantage of IDLVQ-C can be attributed to the adaptive feature extractor, which is tuned in each learning session.

Table 5: Prediction accuracy on CUB base and novel classes using the 10-way 5-shot incremental setting.

| Base classes | sessions | | | | | | | | | | |
| --- | --- | --- | --- | --- | --- | --- | --- | --- | --- | --- | --- |
| | 1 | 2 | 3 | 4 | 5 | 6 | 7 | 8 | 9 | 10 | 11 |
| Fine-tune | 77.30 | 44.23 | 36.28 | 27.52 | 25.96 | 23.05 | 17.68 | 13.07 | 11.78 | 10.99 | 10.71 |
| Joint train | 77.30 | 75.83 | 75.25 | 74.51 | 74.58 | 73.74 | 73.95 | 73.25 | 73.11 | 73.25 | 73.18 |
| iCaRL (Rebuffi et al., 2017) | 77.30 | 59.38 | 58.81 | 54.43 | 48.27 | 43.28 | 39.17 | 34.91 | 32.43 | 29.36 | 25.87 |
| Rebalancing (Hou et al., 2019) | 77.30 | 66.43 | 60.32 | 55.36 | 46.39 | 41.76 | 37.12 | 32.58 | 31.26 | 27.03 | 24.25 |
| ProtoNet (Snell et al., 2017) | 77.30 | 72.55 | 72.21 | 72.06 | 71.64 | 71.29 | 71.02 | 70.94 | 70.67 | 70.60 | 70.53 |
| ILVQ (Xu et al., 2012) | 77.30 | 74.18 | 73.57 | 72.66 | 72.56 | 71.57 | 71.14 | 71.12 | 71.02 | 70.98 | 70.85 |
| SDC (Yu et al., 2020) | 77.34 | 76.05 | 75.21 | 74.12 | 72.36 | 71.81 | 71.68 | 71.43 | 71.25 | 71.27 | 70.96 |
| Imprint (Qi et al., 2018) | 77.02 | 74.76 | 74.57 | 73.69 | 72.69 | 70.88 | 70.34 | 70.12 | 70.07 | 69.84 | 69.27 |
| **IDLVQ-C** | **77.37** | **76.32** | **75.90** | **75.91** | **75.49** | **74.86** | **74.58** | **74.37** | **74.02** | **73.39** | **73.32** |
| Novel classes | sessions | | | | | | | | | | |
| | 1 | 2 | 3 | 4 | 5 | 6 | 7 | 8 | 9 | 10 | 11 |
| Fine-tune | - | **66.23** | 26.86 | 18.12 | 16.16 | 15.85 | 13.76 | 13.68 | 12.23 | 11.37 | 10.81 |
| Joint train | - | 47.78 | 36.55 | 34.77 | 36.68 | 38.52 | 38.91 | 40.37 | **41.50** | **42.51** | **43.28** |
| iCaRL (Rebuffi et al., 2017) | - | 35.18 | 33.97 | 27.04 | 21.99 | 23.99 | 23.04 | 24.01 | 22.96 | 24.04 | 24.59 |
| Rebalancing (Hou et al., 2019) | - | 45.53 | 35.24 | 20.39 | 20.25 | 19.65 | 20.91 | 21.29 | 21.95 | 21.60 | 22.97 |
| ProtoNet (Snell et al., 2017) | - | 41.86 | 35.01 | 29.72 | 29.43 | 28.72 | 28.65 | 30.02 | 29.56 | 30.59 | 30.19 |
| ILVQ (Xu et al., 2012) | - | 40.74 | 32.89 | 29.54 | 29.30 | 30.38 | 28.82 | 32.12 | 29.17 | 30.32 | 30.09 |
| SDC (Yu et al., 2020) | - | 58.45 | 40.65 | 35.77 | 33.23 | 31.16 | 30.24 | 33.25 | 30.89 | 32.26 | 32.08 |
| Imprint (Qi et al., 2018) | - | **59.69** | **44.15** | **38.68** | 38.11 | 40.46 | 41.14 | 42.17 | **40.33** | 40.47 | 40.37 |
| **IDLVQ-C** | - | 58.72 | 42.18 | 37.86 | **39.97** | **40.84** | **41.30** | **43.21** | 40.32 | **42.34** | **42.30** |

The test accuracy on CUB dataset using 10-way 10-shot and 10-way 20-shot incremental settings are reported in Table 7 and 8, respectively. The prediction accuracy improves in all methods with more training samples per class. The iCaRL and Rebalancing methods show the most significant improvement when the number of training samples increases. The proposed IDLVQ-C is effective in different incremental few-shot scenarios as it achieves the best performance on 5-shot, 10-shot and 20-shot settings.

## A.5 Experiment details for incremental few-shot regression

**Sinusoidal wave**: A six-layer feedforward neural network with ReLU nonlinear activation is used as the feature extractor. IDLVQ-R learns 10 reference vectors and targets from the base task. In

Table 6: Prediction accuracy on *mini*ImageNet base and novel classes using the 5-way 5-shot incremental setting.

| Base classes | sessions | | | | | | | | |
|---|---|---|---|---|---|---|---|---|---|
| | 1 | 2 | 3 | 4 | 5 | 6 | 7 | 8 | 9 |
| Fine-tune | 64.25 | 32.28 | 20.87 | 6.95 | 3.17 | 3.16 | 1.92 | 1.53 | 1.46 |
| Joint train | 64.25 | 63.30 | 62.83 | 62.16 | 62.18 | **62.68** | 61.86 | **61.87** | **61.89** |
| iCaRL (Rebuffi et al., 2017) | 64.25 | 51.66 | 48.97 | 45.62 | 37.39 | 30.86 | 28.68 | 26.83 | 24.47 |
| Rebalancing (Hou et al., 2019) | 64.25 | 52.87 | 50.16 | 44.78 | 37.48 | 28.75 | 25.58 | 22.97 | 21.57 |
| ProtoNet (Snell et al., 2017) | 64.25 | 59.27 | 58.88 | 58.69 | 58.22 | 57.63 | 57.03 | 56.80 | 56.47 |
| ILVQ (Xu et al., 2012) | 64.25 | 60.24 | 59.62 | 59.02 | 58.61 | 57.71 | 57.16 | 56.83 | 56.49 |
| SDC (Yu et al., 2020) | 64.62 | 63.58 | 62.78 | 61.12 | 60.29 | 59.37 | 59.05 | 59.97 | 59.87 |
| Imprint (Qi et al., 2018) | 64.71 | 63.52 | 62.96 | 62.13 | 61.17 | 61.27 | 60.63 | 59.86 | 59.64 |
| **IDLVQ-C** | **64.77** | **63.77** | **63.22** | **62.44** | **61.22** | 61.47 | **60.97** | 60.66 | 60.44 |

| Novel classes | sessions | | | | | | | | |
|---|---|---|---|---|---|---|---|---|---|
| | 1 | 2 | 3 | 4 | 5 | 6 | 7 | 8 | 9 |
| Fine-tune | - | 4.07 | 4.49 | 3.75 | 1.93 | 1.53 | 1.77 | 1.61 | 1.36 |
| Joint train | - | 4.80 | 9.84 | 13.26 | 12.30 | 14.03 | **14.01** | **15.21** | **15.62** |
| iCaRL (Rebuffi et al., 2017) | - | 4.60 | 8.09 | 8.92 | 7.87 | 9.10 | 8.19 | 7.86 | 7.70 |
| Rebalancing (Hou et al., 2019) | - | 5.29 | 8.23 | 9.43 | 8.00 | 8.93 | 8.81 | 8.83 | 8.27 |
| ProtoNet (Snell et al., 2017) | - | 5.32 | 8.41 | 9.79 | 11.42 | 12.14 | 11.67 | 11.39 | 11.35 |
| ILVQ (Xu et al., 2012) | - | 5.25 | 9.29 | 10.47 | 12.09 | 12.35 | 11.86 | 11.45 | 11.34 |
| SDC (Yu et al., 2020) | - | 12.23 | 11.05 | 10.12 | 12.33 | 11.46 | 10.81 | 12.58 | 13.30 |
| Imprint (Qi et al., 2018) | - | **15.81** | **12.21** | **13.83** | **16.09** | 13.81 | 12.45 | 12.93 | 13.69 |
| **IDLVQ-C** | - | 13.07 | 12.19 | 13.34 | 15.86 | **14.14** | 12.55 | 13.11 | 13.94 |

Table 7: Prediction accuracy on CUB using the 10-way 10-shot incremental setting.

| Method | sessions | | | | | | | | | | |
|---|---|---|---|---|---|---|---|---|---|---|---|
| | 1 | 2 | 3 | 4 | 5 | 6 | 7 | 8 | 9 | 10 | 11 |
| Fine-tune | 77.30 | 47.16 | 36.34 | 26.92 | 24.08 | 21.24 | 17.19 | 14.31 | 12.73 | 11.75 | 11.43 |
| Joint train | 77.30 | 73.29 | **71.54** | **68.72** | **66.38** | **65.42** | **64.98** | **65.74** | **64.82** | **64.47** | **64.16** |
| iCaRL (Rebuffi et al., 2017) | 77.30 | 59.66 | 56.24 | 52.26 | 48.77 | 46.37 | 44.54 | 43.17 | 42.35 | 41.19 | 40.92 |
| Rebalancing (Hou et al., 2019) | 77.30 | 64.53 | 58.35 | 53.82 | 49.27 | 47.12 | 45.16 | 43.05 | 42.37 | 41.02 | 40.86 |
| ProtoNet (Snell et al., 2017) | 77.30 | 69.82 | 66.12 | 63.19 | 61.17 | 58.85 | 58.04 | 57.75 | 55.84 | 55.82 | 55.60 |
| ILVQ (Xu et al., 2012) | 77.30 | 71.26 | 66.84 | 63.82 | 62.66 | 59.71 | 58.92 | 58.11 | 56.31 | 56.14 | 56.03 |
| SDC (Yu et al., 2020) | 77.34 | 74.67 | 69.73 | 66.71 | 66.49 | 62.14 | 61.33 | 59.84 | 58.01 | 57.39 | 56.62 |
| Imprint (Qi et al., 2018) | 77.02 | 74.07 | 70.26 | 66.84 | 64.45 | 62.46 | 61.85 | 61.02 | 59.20 | 58.93 | 58.43 |
| **IDLVQ-C** | **77.37** | **74.79** | 70.96 | 68.08 | 65.94 | 64.12 | 63.58 | 62.98 | 60.85 | 60.54 | 59.72 |

Table 8: Prediction accuracy on CUB using the 10-way 20-shot incremental setting.

| Method | sessions | | | | | | | | | | |
|---|---|---|---|---|---|---|---|---|---|---|---|
| | 1 | 2 | 3 | 4 | 5 | 6 | 7 | 8 | 9 | 10 | 11 |
| Fine-tune | 77.30 | 47.68 | 36.75 | 27.01 | 24.56 | 21.72 | 17.88 | 15.24 | 12.84 | 11.89 | 11.52 |
| Joint train | 77.30 | 74.34 | **72.59** | **70.85** | **70.14** | **70.01** | **69.69** | **69.67** | **69.48** | **69.41** | **69.42** |
| iCaRL (Rebuffi et al., 2017) | 77.30 | 68.85 | 63.56 | 60.34 | 57.71 | 56.28 | 55.97 | 55.02 | 54.62 | 52.21 | 52.23 |
| Rebalancing (Hou et al., 2019) | 77.30 | 69.36 | 65.49 | 61.32 | 59.30 | 58.68 | 58.77 | 57.42 | 56.25 | 55.33 | 54.17 |
| ProtoNet (Snell et al., 2017) | 77.30 | 70.84 | 67.82 | 64.81 | 62.95 | 61.71 | 60.98 | 60.73 | 59.51 | 59.30 | 58.95 |
| ILVQ (Xu et al., 2012) | 77.30 | 71.50 | 68.77 | 66.12 | 64.31 | 62.89 | 62.31 | 62.00 | 61.52 | 59.79 | 59.23 |
| SDC (Yu et al., 2020) | 77.34 | 74.62 | 71.63 | 68.66 | 66.75 | 65.24 | 64.21 | 63.62 | 61.97 | 61.54 | 61.11 |
| Imprint (Qi et al., 2018) | 77.02 | 74.14 | 70.72 | 67.75 | 65.88 | 64.63 | 64.28 | 63.82 | 62.00 | 61.84 | 61.40 |
| **IDLVQ-C** | **77.37** | **74.84** | 72.07 | 69.05 | 67.28 | 65.51 | 65.19 | 64.84 | 62.77 | 62.55 | 61.96 |

each incremental learning sessions, 5 pairs of reference vectors and targets are added. After new reference vectors and targets are fine-tuned, the model is capable of make prediction for all seen tasks. 10 exemplars are selected uniformly from the training set in the base task. In the incremental learning session of the 2nd task, the model is fine-tuned on 10 exemplars and 5 novel training samples. After the training converges, 5 novel training samples in the current task are added to the exemplar set. In the incremental learning session of the 3rd task, the model is fine-tuned with 15 exemplars from old tasks and 5 novel training samples.

**3D spatial data**: We follow the same training and test protocols as the sinusoidal wave dataset. We choose 40, 15 and 15 reference vectors and targets for 1st, 2nd and 3rd tasks, respectively. Adding more reference vectors does not result in obvious improvement in accuracy in our experiments.

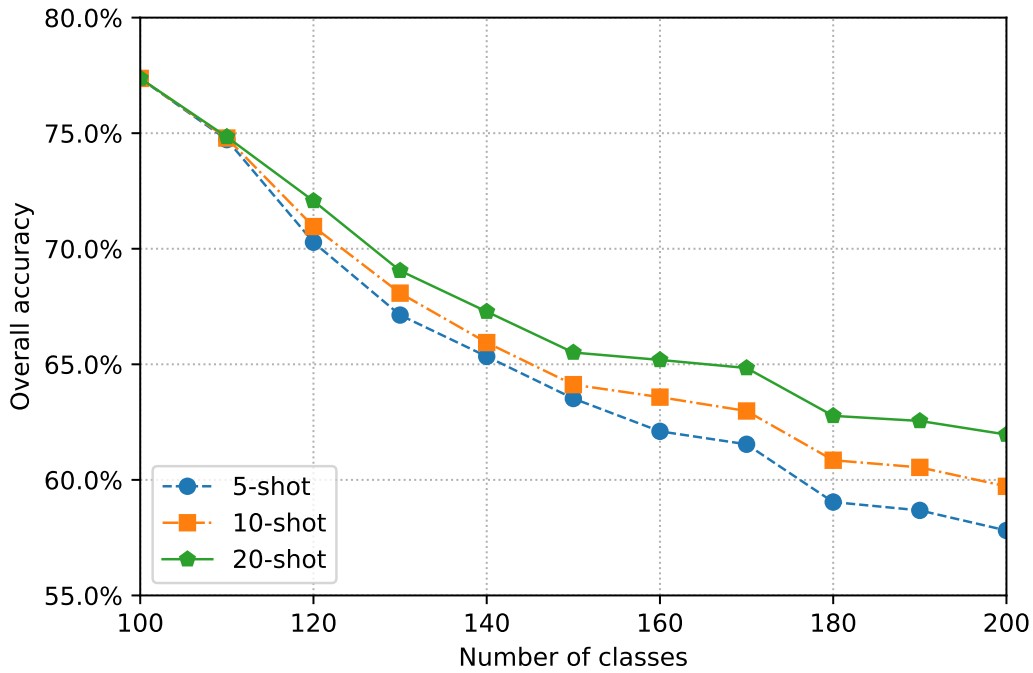

Figure 2: Comparison results of different few-shot settings, evaluated with ResNet18 on CUB dataset

## A.6    VISUALIZATION OF IDLVQ-C

We show the visualization of standard neural networks and IDLVQ-C with/without intra-class variation loss in Fig. 3. MNIST dataset is used as a toy example for visualization. Classes 0-7 are old classes with sufficient training samples, and classes 8 and 9 are novel classes with few-shot training samples. It can be observed in Fig. 3(a) and 3(b) that standard neural networks and IDLVQ-C (without intra-class variation loss) trained by cross-entropy loss do not have compact intra-class variation. Consequently, features of novel classes are more likely to overlap with old classes. In this case, the performance of class-incremental learning degrades very quickly because the classifier cannot distinguish between features from different classes. In comparison, the proposed IDLVQ-C makes intra-class variation compact and leaves large margin between classes in Fig 3(c). As a result, the features of novel classes are less likely to overlap existing classes. The compact intra-class variation and large margin between classes make features of novel classes distinguishable so that learning novel classes is easier. In addition, the margin based loss only updates the model parameters when necessary and avoids catastrophic forgetting of old classes.

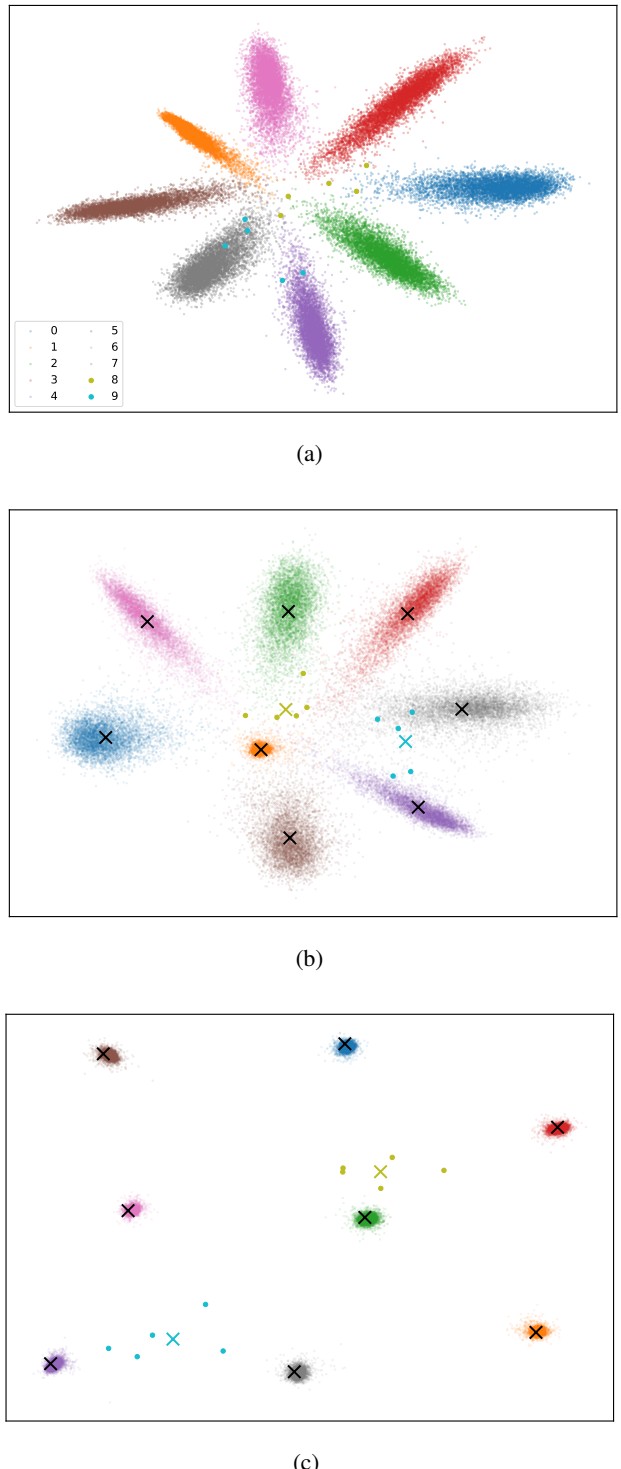

Figure 3: Visualization of feature spaces in different methods. Dots represent features of samples and crosses denote references vectors of classes. (a) standard neural networks; (b) IDLVQ-C without $\mathcal{L}_{intra}$; (c) IDLVQ-C.

