# OpenReview forum: "Incremental few-shot learning via vector quantization in deep embedded space"
_ICLR.cc/2021/Conference — ICLR 2021 Poster_

### Official Review · AnonReviewer4 · 2020-10-27
**a paper with originality but with some limitations too**

**Rating:** 5
**Confidence:** 3

**Review:**

This paper suggests to use a generative model to address the problem of 'few shot' incremental learning. The idea is to classify input data by maintaining a population of prototypes and measuring the distance of the examples to be classified from these prototypes. As, each prototype represents a class, an example to be classified is assigned to the class of the closest prototype.  The learning of the neural network transforming input data to the prototype space is learned using 2 loss functions: the first one maximizes the margin between the distance to the prototype of the correct class and the other prototypes, the second one makes the clusters as compact as possible.

The method is experimentally validated on 2 tasks: an incremental image classification task and an incremental regression task.

The use of generative methods seems a good thing to limit catastrophic forgetting, by anchoring classes on prototypes.  I am not aware of any other work that has proposed this before, which gives a certain originality to the paper. In addition, I found the paper clearly written.

On the negative side, I'm concerned by 3 points. My first concern is related to the size of the training sets. The paper addresses the very particular case where the number of training data per problem is very low, a case where classical incremental methods do not work well. On the other hand, the paper says nothing about the behavior of the method when the size increases.  Figure 2 shows that the performance increases, but the question is whether it still works better than traditional methods in such cases. If it does not, this would reduce the scope of the method considerably.

My second concern is that the authors seem to completely ignore the recent literature on GMM + neural networks. It is a literature that I myself know little about, but I know that it is substantial.  Below are some related references. One should confront the proposed method with this literature and not simply say, as said in section 3.2, that this topic has not attracted much attention in the community.
- J. Snell, K. Swersky, and R. Zemel. Prototypical networks for few-shot learning. In NeurIPS, 2017.
- A Gaussian Mixture Model layer jointly optimized with discriminative features within a Deep Neural Network architecture, Ehsan Variani; Erik McDermott; Georg Heigold,  2015 IEEE International Conference on Acoustics, Speech and Signal Processing (ICASSP)
- Prototype discriminative learning for face image set classification, W Wang, R Wang, S Shan, X Chen - Asian Conference on Computer Vision 2016
- Simultaneous learning of reduced prototypes and local metric for image set classification, Z Ren, B Wu, Q Sun, M Wu, Expert Systems with Applications Volume 134, 15 November 2019
- Joint prototype and metric learning for set-to-set matching: Application to biometrics
M Leng, P Moutafis, IA Kakadiaris 2015 IEEE 7th International Conference on Biometrics Theory, Applications ...

Finally, I did not find the experimental validation very convincing, as far as the part on incremental learning is concerned. The proposed method is only slightly better than methods that are not at the state of the art level. In this regard, I also noted that recent incremental learning methods were not cited in the state of the art. I invite the authors to look at the papers:
- "Prabhu et. al., Gdumb: A simple approach that questions our progress in continual learning.  ECCV2020"
- Zhao, Bowen, Xi Xiao, Guojun Gan, Bin Zhang, et Shu-Tao Xia. « Maintaining Discrimination and Fairness in Class Incremental Learning ». In 2020 IEEE/CVF Conference on Computer Vision and Pattern Recognition (CVPR),
- Tao, Xiaoyu, Xinyuan Chang, Xiaopeng Hong, Xing Wei, et Yihong Gong. « Topology-Preserving Class-Incremental Learning ». In ECCV, 16, 2020.
- Hou, Saihui, Xinyu Pan, Chen Change Loy, Zilei Wang, et Dahua Lin. « Learning a Unified Classifier Incrementally via Rebalancing ». In 2019 IEEE/CVF Conference on Computer Vision and Pattern Recognition (CVPR),
which can provide both recent baselines and also entry points to additional recent literature.

---

> ### Author Response · Authors · 2020-11-22
> **Response to Reviewer#4**
>
> We appreciate the reviewer for the insightful and constructive comments.
>
> **Comment**: On the other hand, the paper says nothing about the behavior of the method when the size increases. Figure 2 shows that the performance increases, but the question is whether it still works better than traditional methods in such cases. If it does not, this would reduce the scope of the method considerably.
>
> **Answer**: We appreciate the reviewer to point out the lack of discussion about the case of different numbers of training samples per class in new tasks. We add the full results of 10-way 10-shot and 10-way 20-shot results on CUB dataset in Table 7 and 8. Based on the result, our method is still very effective when more training samples per class are available. We do not experiment more training samples per class because it is beyond the scope of few-shot scenarios.
>
> **Comment**: My second concern is that the authors seem to completely ignore the recent literature on GMM + neural networks. It is a literature that I myself know little about, but I know that it is substantial. Below are some related references. One should confront the proposed method with this literature and not simply say, as said in section 3.2, that this topic has not attracted much attention in the community.
>
> **Answer**: We appreciate the reviewer to point out the lack of review of recent advances of GMM + NN (Prototype + NN). We include relevant papers in our revised version. We did cite 6 papers in section 3.2 related to prototype + NN and ProtoNet by Snell et al. is reimplemented in the experiment section in our original submission. Due to the limit of 8 pages, we only mention papers that are directly related incremental learning and few-shot learning in the original submission.
>
> **Comment**:  I did not find the experimental validation very convincing, as far as the part on incremental learning is concerned. The proposed method is only slightly better than methods that are not at the state of the art level. In this regard, I also noted that recent incremental learning methods were not cited in the state of the art. I invite the authors to look at the papers.
>
> **Answer**: We appreciate the reviewer to point out some recent state-of-the-art papers in incremental learning. We review additional papers in related methods (Hou et al. 2019CVPR is already reviewed in original submission). In addition, we compare with Hou et al. 2019CVPR in the revised version and report the results in Table 1, 2, 5, 6, 7 and 8. We did compare to a very strong baseline SDC (Yu et al., 2020CVPR) in our original submission. We notice that ECCV2020 was held between Aug 23-28, 2020, which is less than two months to the paper submission deadline of ICLR 2021. According to ICLR policy, ECCV2020 papers are considered as concurrent and authors are not required to review and compare with concurrent papers. We do not compare with Zhao et al, 2020CVPR because their code is not released.

---

### Official Review · AnonReviewer2 · 2020-10-28
**Interesting extention of prototype based approaches for incremental few-shot learning**

**Rating:** 6
**Confidence:** 3

**Review:**

This paper presents a new non-parametric method for few-shot incremental learning. The aim is to perform few-shots classification and regression while being robust to catastrophic forgetting when trying to learn new classes.
The method learns exemplars (reference vectors) and feature representation in order to be able to generalize to unseen classes.

Quality & Clarity:
The paper is well written and the proposed approach seems to work well for few-shot incremental learning. There are some minor issues with presentation and experimental results.

Originality & Significance:
The approach seems a good extension of incremental few-shot learning based on prototypes. Results show that the proposed extension leads to better performance on few-shot classification.

Pros:
- The paper is well presented and motivated
- Results show that the method is performing better than previous approaches on two datasets.
- The ablation study shows the importance of every component in the method

Cons:
- Some statements should be better justified
- Some recent methods haven't been included in the comparison (e.g. Tao et al., 2020). Is there a clear reason?
- The incremental few-shot regression is evaluted on artificial toy problems: Sinusoidal Wave and 3D spatial data

Additional Comments:
- Bottom first page, it is not explained why incremental learning strategies cannot be extended to regression problems easily.
- Top second page, it is not clear why over-fitting exacerbate the difficulty of learning new classes.
- How does the method compare with previous approaches in terms of computational cost?

---

> ### Author Response · Authors · 2020-11-22
> **Response to Reviewer#2**
>
> We appreciate the reviewer for the insightful and constructive comments.
>
> **Comment**: Some recent methods haven't been included in the comparison (e.g. Tao et al., 2020). Is there a clear reason?
>
> **Answer**: We hoped to compare our methods with the TOPIC in Tao et al., 2020, but the implementation on their official GitHub was (and still is) incomplete. The core algorithm part is not released due to “commercial frozen period” based on their post on GitHub. After we re-implement ProtoNet and Weight Imprint, we found ProtoNet and Weight Imprint methods performing much better than the reported results of TOPIC in Tao et al. 2020 under the same experiment settings. We therefore decided to compare against only ProtoNet and Weight Imprint.
>
> **Comment**: The incremental few-shot regression is evaluated on artificial toy problems: Sinusoidal Wave and 3D spatial data
>
> **Answer**: As of today there has not been any existing models nor public data set for incremental few-shot regression. Hence, we have no choice but to create our own data sets for regression task. Note that 3D spatial data, the second case study for regression, are real (that is, not synthetic) albeit small.
>
> **Comment**: Bottom first page, it is not explained why incremental learning strategies cannot be extended to regression problems easily.
>
> **Answer**: We add explanation for the reason why existing methods for class-incremental learning cannot be extended to regression in the bottom of Page 1.  “ In class-incremental learning, the model has to expand output dimensions to learn N' novel classes while keeping the knowledge of existing N classes. Parametric models estimate additional classification weights for novel classes, while nonparametric methods compute the class centroids for novel classes. In comparison, output dimensions in regression problems do not change in incremental learning as neither additional weights nor class centroids are applicable to regression problems. “
>
> **Comment**: Top second page, it is not clear why over-fitting exacerbate the difficulty of learning new classes.
>
> **Answer**: We realized that the sentences in top second page (original submission) could be confusing. We rephrase it as follows:
> “First, the model is biased towards new classes and forgets old classes because the model is fine-tuned on new data only. Meanwhile, the prediction accuracy on novel classes is not good due to over-fitting on few-shot training samples.”
>
> **Comment**: How does the method compare with previous approaches in terms of computational cost?
>
> **Answer**: The total parameter size is very similar to the standard neural networks. The computation cost is very close to iCaRL and SDC, which fine-tunes the entire model. The computation cost of our method is higher than ProtoNet and Imprint, because they only recompute the new parameters in the last layer.

---

### Official Review · AnonReviewer3 · 2020-10-29
**The paper needs more comparison experiments**

**Rating:** 6
**Confidence:** 4

**Review:**

This paper proposes a new method for incremental few-shot learning based on feature quantization. Each class is represented by a reference vector, which is initialized as a centroid of features within that class and fine-tuned using a margin loss, an intra-class loss, and a forgetting loss.

Pros:
+ The paper is overall well-written.
+ The proposed method compares favorably against various baselines, including strong ones such as SDC.
+ Experiments are conducted on both classification and regression datasets.

Cons:
- This paper is very related to the work of Xu et al. [1] but fails to cite it. Both this paper and Xu et al. propose an incremental learning method based on vector quantization and compare it with prototype-based classifiers. Compared with Xu et al., this paper performs VQ learning a deep embedding space and adopts a different learning rule. The relationship with Xu et al. needs to be thoroughly discussed. A baseline approach that naively combines Xu et al. with a deep network by replacing the nearest neighbor classifier in ProtoNet with the VQ classifier by Xu et al., should also be implemented and compared.

- In the classification experiments, the paper only reports the average accuracy for all classes the model has been trained on. It would be helpful to separately report the accuracy of previous classes and new classes so that readers can understand whether
- The margin-based loss is not evaluated in the ablation study.

Minor problems that do not affect my score:
P1: are compressed -> is compressed
P5: repeated “such as”

Overall, I lean towards rejection, mainly because of the lack of discussion and comparison with Xu et al. [1]. I am willing to increase my score if this issue can be fixed in the rebuttal.

[1] Xu, Ye, Furao Shen, and Jinxi Zhao. "An incremental learning vector quantization algorithm for pattern classification." Neural Computing and Applications 21.6 (2012): 1205-1215.

======================================================

I've read the rebuttal and the authors have addressed most of my concerns in the revised paper so I raise my score.

---

> ### Author Response · Authors · 2020-11-22
> **Response to Reviewer#3**
>
> We appreciate the reviewer for the insightful and constructive comments, and pointing out our typos in the manuscript.
>
> **Comment**: This paper is very related to the work of Xu et al. [1] but fails to cite it. Both this paper and Xu et al. propose an incremental learning method based on vector quantization and compare it with prototype-based classifiers. Compared with Xu et al., this paper performs VQ learning a deep embedding space and adopts a different learning rule. The relationship with Xu et al. needs to be thoroughly discussed. A baseline approach that naively combines Xu et al. with a deep network by replacing the nearest neighbor classifier in ProtoNet with the VQ classifier by Xu et al., should also be implemented and compared.
>
> **Answer**: We appreciate the reviewer to point out the lack of review on Xu et al.. The difference between our paper and Xu et al. is summarized below.
> 1.	In Xu et al., the prototype learning is based on some predefined rules (e.g. when to add prototypes, how to update existing prototypes). They also maintain the link of prototypes to help define the updating rules. In our paper, we use a differentiable loss function to learn the prototypes. Our method combines prototype learning with deep feature extractor in an end-to-end learning fashion, while the method in Xu et al does not use gradient based learning.
> 2.	In Xu et al., different classes have different numbers of prototypes and the number of prototypes per class is automatically determined. They automatically assign some prototypes close to the boarder of a class. In our method, we use one reference vector per class. Since each class is well separated from each other and has compact intra-class variation, one reference vector per class is sufficient in our method.
>
> We use ILVQ from Xu et al. on the features from neural networks, and report the results in Table 1, 2, 5, 6, 7 and 8 in the revised version.
>
> **Comment**: In the classification experiments, the paper only reports the average accuracy for all classes the model has been trained on. It would be helpful to separately report the accuracy of previous classes and new classes so that readers can understand whether
>
> **Answer**: We add accuracy for both old and new classes in Table 5 and 6 in the revised version.
>
> **Comment**: The margin-based loss is not evaluated in the ablation study.
>
> **Answer**: We add the ablation study for margin loss in Table 3 in the revised version.

---

### Official Review · AnonReviewer1 · 2020-11-02
**This paper is hard to understand. The experiments do not demonstrate the effectiveness of the proposed method sufficiently.**

**Rating:** 5
**Confidence:** 4

**Review:**

Paper Summary:

This paper proposes a nonparametric method in deep embedded space to address incremental few-shot learning problems. By compressing the learned tasks into a small number of reference vectors, the method could add more reference vectors to the model for each novel task, which could alleviate catastrophic forgetting and improve the performance of related tasks. Finally, this paper evaluates the proposed method on the classification and regression problem, respectively.


Strengths:

The idea of employing reference vectors to address incremental few-shot learning is novel and interesting.


Weakness:

1. The writing of this paper is not clear. The authors do not introduce their motivation clearly. For example, in the Introduction Section, the authors said "To the best of our knowledge, the majority of incremental learning methodologies focus on classification problems and they cannot be extended to regression problems easily." Here, the authors should further interpret the reason for this scenario. Meanwhile, compared with classification, the authors should analyze the challenges for the incremental few-shot regression problem. Besides, the authors should interpret the advantages of employing nonparametric method.

2. The method section is hard to understand. I have no idea of the motivation of the proposed method. The authors should give more interpretations of the proposed method. In Section 4.2, the authors indicate the cross-entropy loss does not guarantee compact intra-class variation in the feature space. The authors should give some visualization analyses to demonstrate their opinion. Meanwhile, the authors should analyze the reason. Eq. (2) and Eq. (4) are two simple loss functions. The authors should give more interpretations of why Eq. (2) and Eq. (4) could guarantee intra-class variation and regularize the drift in the feature space.

3. In Table 1 and 2, for the compared methods, the authors should give detailed citations. Besides, for CUB and miniImagNet dataset, the authors only evaluate their method under the 10-way 5-shot and 5-way 5-shot case. To demonstrate the effectiveness of the proposed method, the authors should evaluate their method on more cases. Based on Table 1 and 2, we are not sure whether the proposed method is effective. Besides, the authors should make a visualization analysis to further demonstrate the effectiveness of the proposed method. Finally, the proposed method contains some hyper-parameters. The authors should analyze the impact of these hyper-parameters.

Overall, though the idea of employing reference vectors is interesting, the motivation of this paper is not clear. The method section is hard to understand. Meanwhile, the experimental results do not demonstrate the effectiveness of the proposed method sufficiently. The authors do not make a visualization analysis of the proposed method. Thus, I think this paper is below the acceptance.

EDIT: The authors' rebuttals have solved my concerns partially. However, there still exist some concerns about this paper.

1. In the introduction, the authors indicate that nonparametric methods compute the class centroids for novel classes. However, I am not clear whether computing class centroids is the only mechanism to solve class-incremental learning. Besides, I recommend citing more papers about this interpretation.

2. The category centers are usually affected by the number of samples. When the samples are scarce, the constructed centers are not accurate, which affects the performance. The authors should give more discussions about category centers.

3. For the regression problem, it is better to give more implementation details and ablation analyses. Besides, whether MSE is the only loss to train regression network.

Based on these concerns, I update the score to 5.

---

> ### Author Response · Authors · 2020-11-22
> **Response to Reviewer#1**
>
> We appreciate the reviewer for the insightful and constructive comments.
>
> **Comment**: …., the authors said "To the best of our knowledge, the majority of incremental learning methodologies focus on classification problems and they cannot be extended to regression problems easily." Here, the authors should further interpret the reason for this scenario.
>
> **Answer**: We add explanation why existing methods for class-incremental learning cannot be extended to regression in the bottom of Page 1, which is copied below for convenience: “In class-incremental learning, the model has to expand output dimensions to learn N' novel classes while keeping the knowledge of existing N classes. Parametric models estimate additional classification weights for novel classes, while nonparametric methods compute the class centroids for novel classes. In comparison, output dimensions in regression problems do not change in incremental learning as neither additional weights nor class centroids are applicable to regression problems. “
>
> **Comment**: …. compared with classification, the authors should analyze the challenges for the incremental few-shot regression problem.
>
> **Answer**: We discussed the challenges for incremental few-shot learning in Paragraph 2 and 3 on the first page. Those challenges apply to both classification and regression.
>
> **Comment**: Besides, the authors should interpret the advantages of employing nonparametric method.
>
> **Answer**: We explain the advantage of non-parametric methods in the introduction part in the lost paragraph of Introduction. (1) Non-parametric methods avoid the biased weights towards novel classes; (2) A unified framework for incremental classification and regression can be developed using non-parametric methods.
>
> **Comment**: In Table 1 and 2, for the compared methods, the authors should give detailed citations.
>
> **Answer**: We add explicit citations in all tables.
>
> **Comment**: Besides, for CUB and miniImagNet dataset, the authors only evaluate their method under the 10-way 5-shot and 5-way 5-shot case. To demonstrate the effectiveness of the proposed method, the authors should evaluate their method on more cases. Based on Table 1 and 2, we are not sure whether the proposed method is effective.
>
> **Answer**: We add more experiment on different cases (10-shot, 20-shot) in Table 7 and 8. It will demonstrate the effectiveness of the proposed method in different scenarios.
>
> **Comment**: Besides, the authors should make a visualization analysis to further demonstrate the effectiveness of the proposed method. Finally, the proposed method contains some hyper-parameters. The authors should analyze the impact of these hyper-parameters.
>
> **Answer**: We added visualizations in Appendix (A5, last page) of the revised version to demonstrate feature spaces generated by our proposed method and the standard neural network. We show the visualization with/without Eq (2). The proposed method leads to compact intra-class variation, which helps class-incremental learning. We add the ablation study for Eq (4) in Table 3.
>
> **Comment**: The authors should analyze the impact of these hyper-parameters.
>
> **Answer**: We add the discussion of hyperparameters in appendix A2. Our method only contains two hyperparameters: weight for intra-class loss and weight for less forgetting loss. Empirically, larger $\lambda_{intra}$ leads to more compact intra-class variation. However, convergence could be slow if  $\lambda_{intra}$ is too large. In addition, larger $\lambda_F$ results in less forgetting in old classes but makes learning novel classes more difficult.

---

### Author Response · Authors · 2020-11-22
**We appreciate all reviewers for their constructive comments. The revised paper is uploaded.**

We appreciate all reviewers for the insightful and constructive comments.

We revised the paper to address reviewers’ questions and concerns.  The revised version is uploaded. We also made point-to-point response to reviewers’ comments. We are happy to make further response if necessary.

---

### Decision · Program_Chairs · 2021-01-07
**Final Decision**

**Decision:**

Accept (Poster)

**Comment:**

There was quite some variance in opinion on this paper, with some reviewers commenting on problems with clarity and experimental evaluation. The authors rebuttals improved the reviewer opinions slightly. The rebuttal and accompanying revisions are convincing, and the new experimental results are convincing and also very much appreciated. This is one of the first papers taking a comprehensive look at incremental, few-shot classification AND regression. Despite some problems with clarity (which were well-addressed in rebuttal and revisions), the paper is original and presents novel ideas about incremental few-shot learning.

Pros: consideration of both few-shot classification and regression, ablation study well-executed and convincing.

(remaining) Cons: some minor problems with clarity - please take reviewer comments on board when preparing the camera ready version.